# Berberine bridge enzyme-like oxidase-catalysed double bond isomerization acts as the pathway switch in cytochalasin synthesis

Jin-Mei Zhang[1,3], Xuan Liu[1,3], Qian Wei[1], Chuanteng Ma[2], Dehai Li [2] & Yi Zou [1✉]

Cytochalasans (CYTs), as well as their polycyclic (pcCYTs) and polymerized (meCYTs) derivatives, constitute one of the largest families of fungal polyketide-nonribosomal peptide (PK-NRP) hybrid natural products. However, the mechanism of chemical conversion from mono-CYTs (moCYTs) to both pcCYTs and meCYTs remains unknown. Here, we show the first successful example of the reconstitution of the CYT core backbone as well as the whole pathway in a heterologous host. Importantly, we also describe the berberine bridge enzyme (BBE)-like oxidase AspoA, which uses $Glu_{538}$ as a general acid biocatalyst to catalyse an unusual protonation-driven double bond isomerization reaction and acts as a switch to alter the native (for moCYTs) and nonenzymatic (for pcCYTs and meCYTs) pathways to synthesize aspochalasin family compounds. Our results present an unprecedented function of BBE-like enzymes and highly suggest that the isolated pcCYTs and meCYTs are most likely artificially derived products.

[1] College of Pharmaceutical Sciences, Southwest University, Chongqing 400715, China. [2] Key Laboratory of Marine Drugs, Chinese Ministry of Education, School of Medicine and Pharmacy, Ocean University of China, Qingdao 266003, China. [3]These authors contributed equally: Jin-Mei Zhang, Xuan Liu.
✉email: zouyi31@swu.edu.cn

Cytochalasans (CYTs), one of the largest families (≥400 isolated compounds) of fungal polyketide-nonribosomal peptide (PK-NRP) hybrid natural products, exhibit a wide range of important pharmaceutical and agricultural activities[1]. They contain the common feature of an isoindole core fused to an 11~14-membered macrocyclic framework (Fig. 1). The structural complexity of CYTs is mainly attributed to four variable bio-conversion processes:[2] (1) initial steps mediated by polyketide-nonribosomal peptide synthases (PKS-NRPSs) for core backbone synthesis, which can incorporate diverse types of amino acids (aromatic or aliphatic amino acids) and introduce different modified polyketide chains (Fig. 1a); (2) tailoring steps that are catalysed by numerous distinctive oxidases to form highly oxidised functional groups (Fig. 1b); (3) intermolecular poly-merization steps that are performed in undefined ways, such as the combination of mono-cytochalasans (moCYTs) with other chemical moieties, via Michael addition, Diels-Alder reaction or heterocycloaddition reactions to form the dimerized or trimerized types of mero-cytochalasans (meCYTs, Fig. 1c); and (4) intramolecular C−C or C−O bond linkages that can convert the common macrocycle framework to the polycyclic skeleton (pcCYTs, Fig. 1d), such as the 5/6/6/5/6-fused penta-cyclic ring in aspergillin PZ (1) and its dehydroxylated derivate 2. Therefore, these fantastic transformation reactions towards moCYT scaffolds represent a good learning example to understand the chemical logic of nature during the construction of complex natural products[3], and more importantly, to provide an insightful biomimetic strategy for chemists to synthesize this family of compounds[4–12].

Since the identification of CYT biosynthetic gene clusters (BGCs) from various fungal species, the biosynthetic pathways and the functions of their corresponding enzymes have been well investigated by many groups over the past two decades[3,13–23]. Many significant and unprecedented chemical reactions have also been discovered. For example, cytochrome P450 (CHGG_01243)-catalysed successive C-H oxidation on nonactivated carbons during chaetoglobosin A biosynthesis[15] and the carbonate functional groups synthesized by multifunctional Baeyer-Villiger mono-oxygenase (BVMO, CcsB) during cytochalasin E biosynthesis[16]. According to previous results, a basic frame diagram of CYT biosynthesis has been established;[3] however, two crucial issues remain unsolved to date. (1) Identification of an initial core backbone synthesized by the four-gene conserved cassette (consisting of PKS-NRPS, trans-ER, hydrolase and the Diels-Alderase, Supplementary Fig. 2) which is common to all cyt BGCs. (2) The mechanism of chemical conversion from moCYTs to both pcCYTs and meCYTs.

We carefully analysed previous works on CYT biosynthesis and found the following information. (1) Reconstitution of aromatic amino acid-type cyt BGCs in Aspergillus nidulans and Aspergillus oryzae failed due to unexpected reduction or tailoring steps by unknown enzymes in these two heterologous hosts (Fig. 1e and Supplementary Fig. 3)[14,17,23]. These mismodified products cannot be recognized by the subsequent native enzymes of cyt BGCs. (2) As shown in Fig. 1f, the conversion of moCYTs to both meCYTs and pcCYTs via Michael addition or cycloaddition may occur on the proposed olefin intermediate of CYT scaffolds[24]. (3) In comparison with aliphatic amino acid-type meCYTs and pcCYTs, aromatic ammino acid-type meCYTs and pcCYTs are relatively rare (Fig. 1c, d)[3], which indicates a uniquely derived step during the synthesis of aliphatic amino acid-type CYTs.

Here, we use the aspo cluster of aliphatic amino acid-type cytochalasin compounds (aspochalasans) as the research target and demonstrate that (1) reconstitution of the four-gene con-served cassette (aspoEHBC) of the aspo cluster is successful in the heterologous host A. nidulans and that the corresponding pro-duction compound aspochalasin Z is the core backbone product of the aspo pathway; (2) the BBE-like oxidase AspoA uses $Glu_{538}$ as the general acid biocatalyst to catalyse an unusual protonation-driven double bond isomerization reaction, presenting an unprecedented function of BBE-like enzymes in natural product biosynthesis, and acts as a switch to alter the native (for moCYTs) and nonenzymatic (for pcCYTs and meCYTs) pathways in syntheses of aspochalasin family compounds.

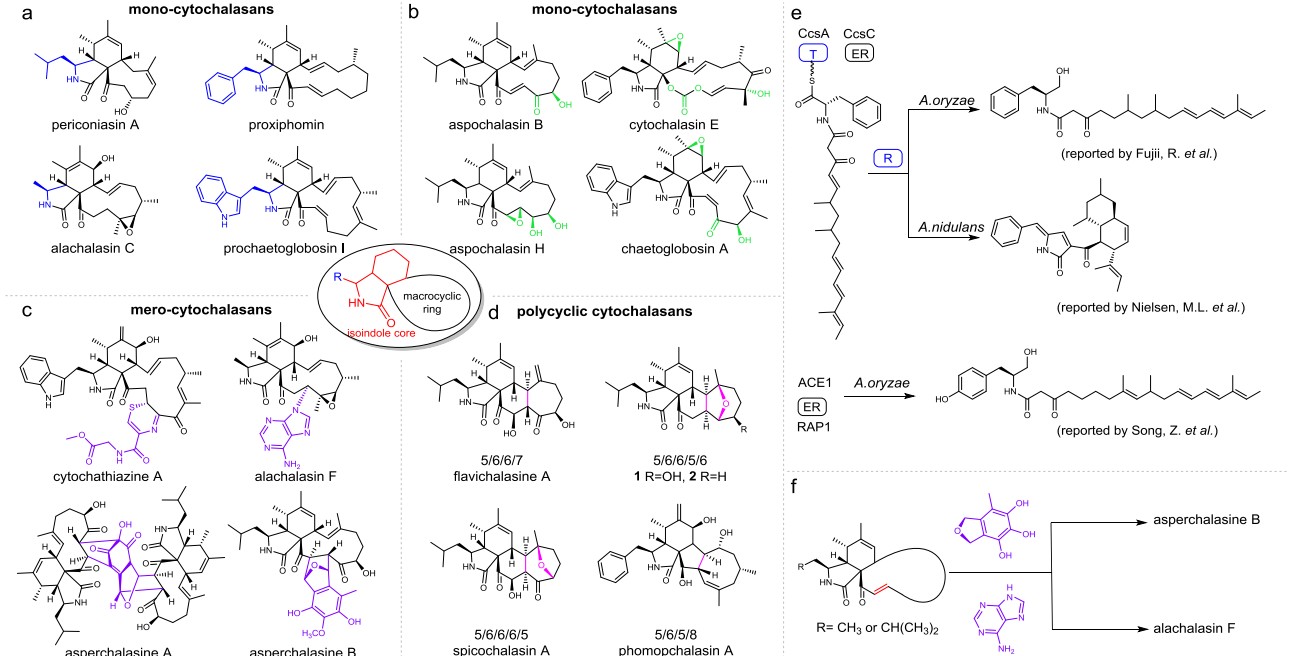

**Fig. 1 Structural and chemical diversity of the cytochalasin family compounds. a–d** Representative mono-, mero- and polycyclic cytochalasans reveal four variable bioconversion processes. **e** Previously unsuccessful examples of the reconstitution of aromatic ammino acid-type cyt BGCs in heterologous hosts. **f** Conversion of moCYTs to both meCYTs and pcCYTs via a proposed olefin intermediate in aliphatic amino acid-type CYT scaffolds.

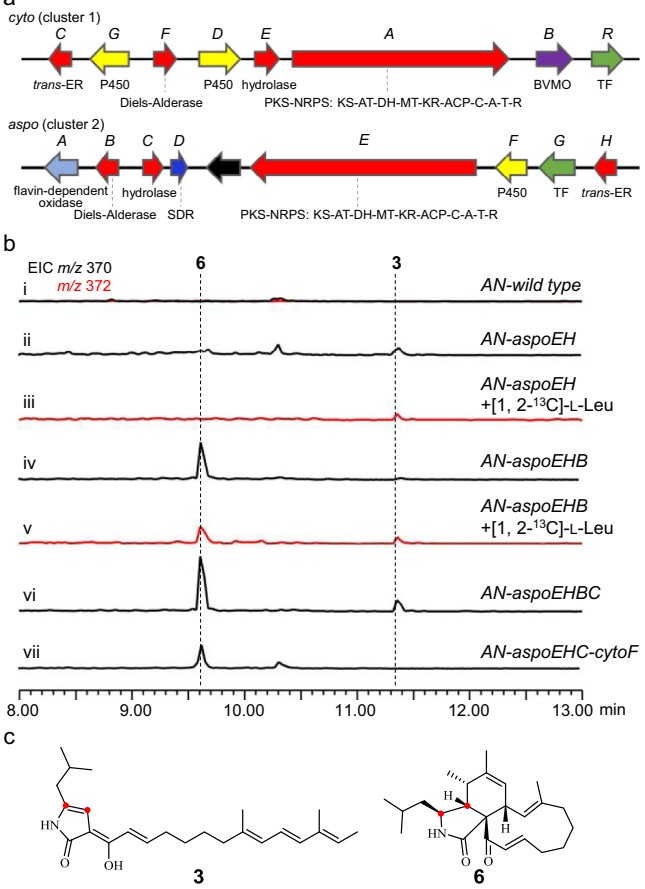

**Fig. 2 Confirmation of the *aspo* cluster and the function of the four-gene conserved cassette. a** Organization and proposed function of the *cyto* and *aspo* clusters in *A. flavipes* KLA03. **b** LC-MS analyses of the culture extracts from the *A. nidulans* transformants. **c** Diagram showing the incorporation of [1,2-$^{13}$C]-L-leucine into **3** and **6**. The extracted ion chromatograms (EICs) were extracted at *m/z* 370 [M + H]$^+$ for **3** and **6**, *m/z* 372 [M + H]$^+$ for **3** and **6** labelled with [1,2-$^{13}$C]-L-leucine.

## Results and discussion

**Bioinformatic analysis revealed two *cyt* BGCs in the fungus *Aspergillus flavipes* KLA03.** Based on the abovementioned facts, to investigate the function of each *cyt* BGC gene, and especially to clarify these two common crucial issues in CYT biosynthesis, we turned our attention to aliphatic amino acid-type CYTs. A special example is *A. flavipes*[20,25–28], the strain that simultaneously produces (1) L-phenylalanine-type moCYTs (Supplementary Fig. 4a) and (2) L-leucine-type moCYT, pcCYT and meCYT aspochalasans (Supplementary Fig. 4b), which indicates a strict regulation rule or a precise polycyclic/polymerization mechanism for aspochalasin conversion.

We sequenced the genome of *A. flavipes* KLA03[25], used CcsA as the probe, and found two possible *cyt* BGCs, which are shown in Fig. 2a. (1) Cluster 1 (*cyto* cluster) shares similar gene compositions and organizations with the *ccs* cluster (Supplementary Fig. 5a), and shares the identical A domain codes of the NRPS module (Supplementary Fig. 5b); thus, cluster 1 should be responsible for the synthesis of L-Phe-type moCYTs. (2) Apart from the four-gene conserved cassette (*aspoEHBC*) and regulation gene (*aspoG*), cluster 2 (*aspo* cluster) has three tailoring genes, a cytochrome P450 monooxygenase gene (*aspoF*), a short chain dehydrogenase/reductase (SDR) gene (*aspoD*) and a flavin-dependent oxidase gene (*aspoA*), where *aspoA* is distinguishable from the flavin-dependent BVMO gene (*cytoB*) in cluster 1. (3)

The coexistence of cluster 1 and cluster 2 highly suggests that the structural diversity of CYTs in *A. flavipes* KLA03 is not due to the promiscuous incorporation of amino acids by the A domain of the NRPS module but rather it is the reason that *A. flavipes* KLA03 harbours one additional *cyt* BGC. Indeed, deletion of the highly homologous PKS-NRPS gene (*ffsA*, Supplementary Fig. 6) in the marine-derived fungus *A. flavipes* CNL-338 abolished the production of aspochalasin-type moCYTs and pcCYTs[20].

**Heterologous expression of the PKS-NRPS gene and *trans*-ER gene in *A. nidulans* led to the production of shunt product.** We next planned to investigate the function of each gene in the *aspo* cluster as well as the corresponding synthetic steps via a gene combination strategy. The PKS-NRPS gene *aspoE* and its *trans*-ER partner *aspoH* were first heterologously expressed in *A. nidulans* (*AN-aspoEH*). After 3 days of solid medium culture followed by extraction with ethyl acetate, a trace amount of compound **3** (~0.25 mg/L) with *m/z* 370 [M + H]$^+$ was produced in *AN-aspoEH* by liquid chromatography-mass spectrometry (LC-MS) analysis (Fig. 2b, i, ii). When 1 mM [1,2-$^{13}$C]-L-leucine was added, the molecular weight of **3** increased by 2 amu (Fig. 2b, iii and Supplementary Fig. 7a), demonstrating that L-leucine is indeed the amino acid component of **3** (Fig. 2c). The molecular weight of **3** is consistent with that of the expected Knoevenagel condensation product **4** (Fig. 3a); however, the main UV absorption peaks of **3** (λ$_{max}$) were located at 274 nm and 386 nm (Supplementary Fig. 7b), which indicates that **3** could be the 1,3-dihydro-2*H*-pyrrol-2-one tautomer rather than the required 1,5-dihydro-2*H*-pyrrol-2-one tautomer **4**. Isolation of **3** from the large-batch fermentation cultures of *AN-aspoEH* was carried out (SI), and its structure was confirmed by NMR analyses (Fig. 3a, Supplementary Table 6 and Supplementary Figs. 44–49, the elucidation process for compound **3** is described in SI). Although we obtained shunt compound **3** rather than the expected product **4** from strain *AN-aspoEH*, possibly due to the rapid tautomerization of **4** to **3** in vivo[29], the production of **3** fully demonstrates that (1) the working programs of both the hrPKS module (for polyketide chain extension) and NRPS module (for polyketide chain transfer and amino acid selection) of AspoE are correct; and (2) under our culture conditions, no enzymes from *A. nidulans* can catalyse the reduction of putative key aldehyde intermediate **5** to yield alcohol product **5'** (Fig. 3a), which is usually observed during the reconstitution of other CYT pathways (Fig. 1e)[14,17].

**The additional introduction of the proposed Diels-Alderase and hydrolase genes into *A. nidulans* successfully reconstituted core backbone synthesis.** The production of **3** in *AN-aspoEH* strongly suggests that the nonenzymatic conversion of the 1,5-dihydro tautomer to the 1,3-dihydro tautomer should be completely inhibited during the actual biosynthetic pathway of aspochalasin (Fig. 3a). Therefore, the Diels-Alder reaction must occur very rapidly, before the nonenzymatic tautomerization reaction to capture the possible Knoevenagel condensation product **4**. Based on this hypothesis, we introduced the proposed Diels-Alderase gene *aspoB* into *AN-aspoEH*, and the resulting strain *AN-aspoEHB* produced another compound **6** (~0.3 mg/L, aspochalasin Z), with *m/z* 370 [M + H]$^+$ (Fig. 2b, iv). The incorporation of [1,2-$^{13}$C]-L-leucine into **6** was also observed (Fig. 2b, c, v and Supplementary Fig. 7). Structural confirmation of **6** by NMR analyses (Fig. 3a and Supplementary Table 7 and Supplementary Figs. 50–56) not only indicated that the cooperation of Diels-Alderase with the PKS-NRPS and *trans*-ER enzymes is important for the interception of the shunt pathway to

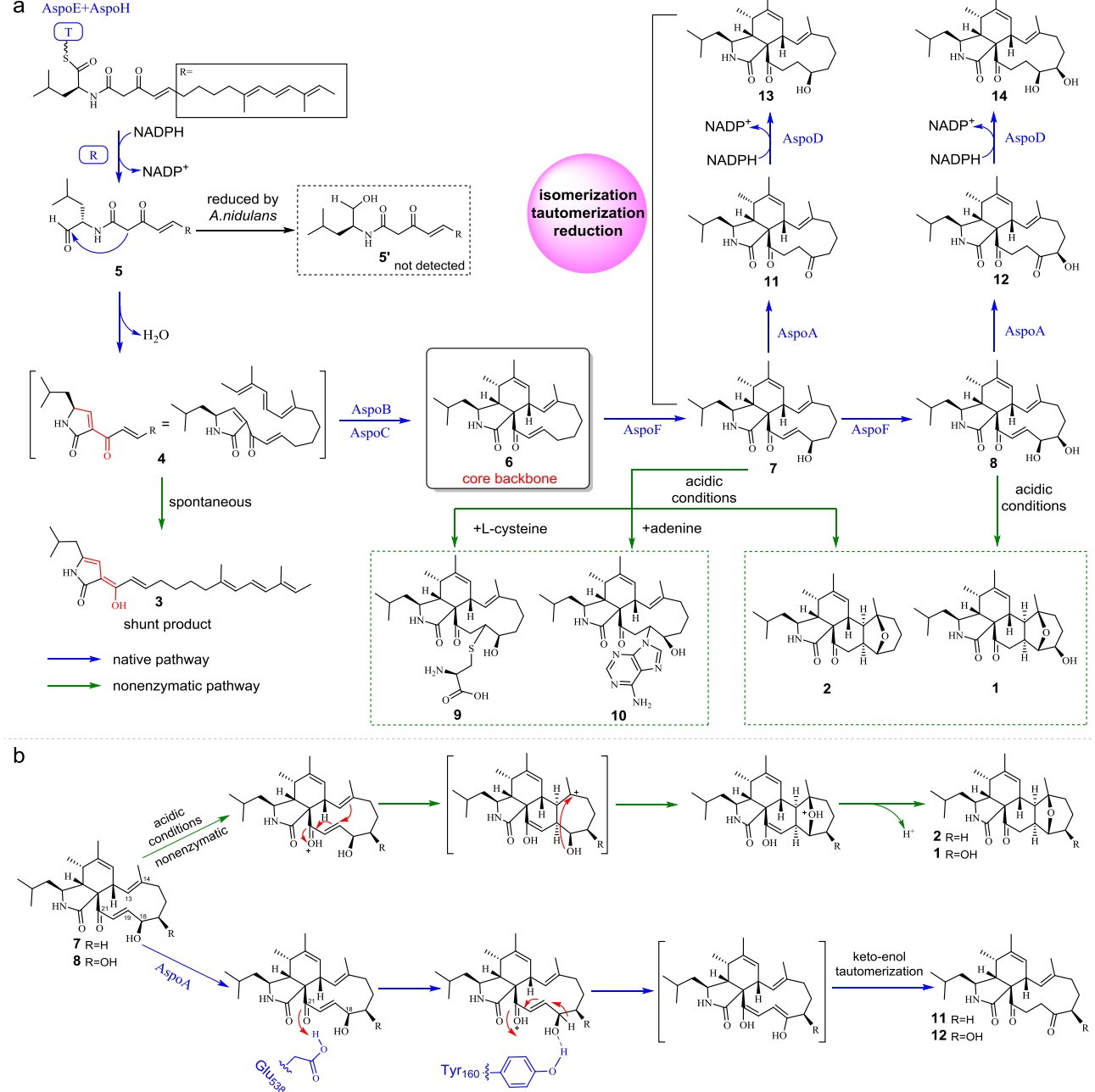

**Fig. 3 The proposed biosynthetic pathway of the aspochalasin family of compounds. a** The pathway identified in this work shows the enzymatic (for the monocyclic) and nonenzymatic (for the mero- and polycyclic) chemical conversions, where **6** is the core backbone and AspoA acts as a pathway switch. **b** The proposed mechanism of conversion of **7** or **8** to **2** or **1** via protonation of the $C_{21}$ carbonyl group under acidic conditions and the proposed mechanism of AspoA-catalysed isomerization of **7** or **8** to **11** or **12** by $Glu_{538}$-mediated protonation of the $C_{21}$ carbonyl group.

**3** but also showed the first successful example of the reconstitution of the CYT scaffold in a heterologous host.

Further addition of the hydrolase gene *aspoC* (AN-*aspoEHBC*) significantly increased the yield of **6** (almost 60%, Fig. 2b, vi). Recently, using synthesized mimic substrates, Zhang et al. proposed that the hydrolase-catalysed reaction might occur prior to the Diels-Alderase-catalysed reaction during pyrichalasin H biosynthesis[29]. Formation of a hydrolase-bound intermediate (via covalent binding to retain the correct tautomer form of the substrate) is crucial for the subsequent Diels-Alder reaction. However, in our case, hydrolase AspoC only influences but does not determine the catalytic ability of AspoB, whereas Diels-Alderase seems to play the principal role. Indeed, exchange of

*aspoB* for *cytoF* (the proposed Diels-Alderase gene in cluster 1) resulted in strain AN-*aspoEHC-cytoF* that retained the ability to produce **6** (Fig. 2b, vii). Therefore, we proposed that the hydrolase AspoC possibly provides a structural cavity (not via covalent binding) to retain **4** in the correct tautomer form to react with Diels-Alderase AspoB during core backbone **6** biosynthesis.

**The pcCYTs and meCYTs are not enzyme-catalysed products from the biosynthetic process of the aspochalasin family of compounds.** Introduction of the cytochrome P450 monooxygenase gene *aspoF* into strain AN-*aspoEHBC* (AN-*aspoEHBCF*) gave two products, **7** (~1.25 mg/L, TMC-196) and

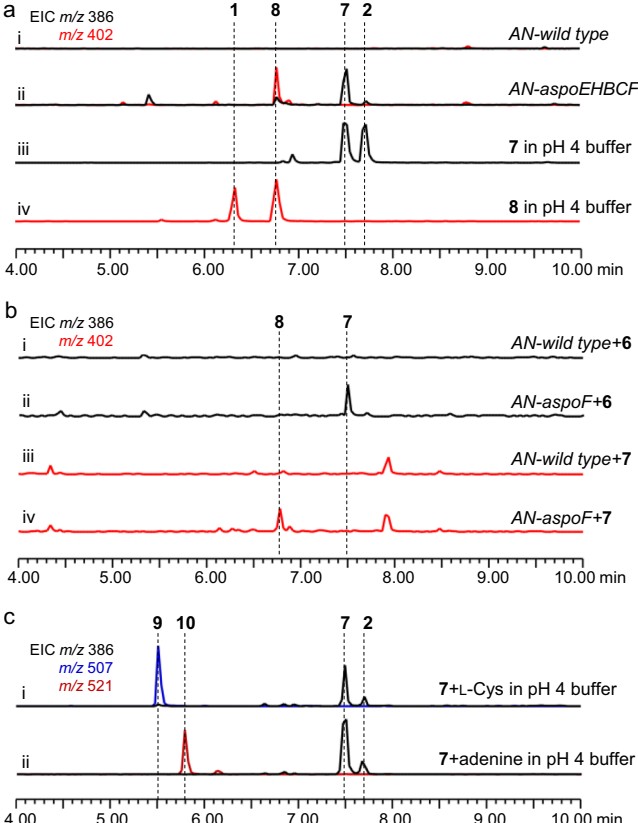

**Fig. 4 Confirmation of the function of the *aspoF* gene and the nonenzymatic conversions towards 7 and 8. a** LC-MS analyses of the culture extracts from the *A. nidulans* transformants and the products from **7** and **8** conversion under acidic conditions. **b** Chemical feedings confirmed that AspoF catalyses only successive hydroxylation reactions to form **7** and **8**. **c** Mimic synthesis of mero-cytochalasans via Michael addition using **7** as the example substrate. The EICs were extracted at $m/z$ 386 $[M + H]^+$ for **7** and **2**, $m/z$ 402 $[M + H]^+$ for **8** and **1**, $m/z$ 507 $[M + H]^+$ for **9**, and $m/z$ 521 $[M + H]^+$ for **10**.

**8** (~1.0 mg/L, aspochalasin D), with $m/z$ 386 $[M + H]^+$ and $m/z$ 402 $[M + H]^+$, respectively (Fig. 4a, i, ii). These two compounds were purified through large-batch fermentation and isolation (SI). When **7** and **8** were dissolved in $CDCl_3$ for NMR analyses, we found that these two compounds were converted to new compounds, **2** and **1**, respectively (Supplementary Fig. 8). We carefully repurified **7** and **8**, accompanied by **1** and **2**, and confirmed their structures by NMR analyses in DMSO-$d_6$ and $CDCl_3$, respectively (Fig. 3a and Supplementary Tables 4, 5, 8, 9 and Supplementary Fig. 30–43, 57–70). The results showed that (1) **7** and **8** are the monohydroxyl and dihydroxyl products of **6**, respectively; however, (2) **1** and **2** contain the complex 5/6/6/5/6-fused pentacyclic system, and they are nonenzymatically derived from **8** and **7**; and (3) in the slightly acidic chloroform environment, the $C_{21}$ carbonyl groups of **7** and **8** might be protonated, which induces new C-C bond formation between $C_{13}$ and $C_{19}$. The obtained carbocation at $C_{14}$ could then finally be quenched by the $C_{18}$ hydroxyl group (Fig. 3b). To verify this hypothesis, we incubated 100 μM **7** and **8** in pH 4 Tris-HCl buffer, and the conversions of **7** and **8** to **2** and **1** were clearly observed after 10 h (Fig. 4a, iii, iv). Moreover, P450 AspoF catalysed only the successive hydroxylation of **6** to **7** and **7** to **8**, confirmed by in vivo feeding (Fig. 4b, i–iv).

According to the above results, pcCYTs **1** and **2** are the nonenzymatic conversion products obtained from simple moCYTs **8** and **7**. This discovery is the opposite of a previous biosynthetic hypothesis, that the formation of polycyclic skeletons in CYTs, from the common macrocycle framework, may need to involve a series of diverse oxidative reactions[3,12]. This nonenzymatic polycyclic transformation might be related to the highly reactive features of the γ-keto-α,β-unsaturated moiety in **7** and **8**, which might also be important for linking the macrocycle framework to other chemical functional groups via a Diels-Alder reaction, heterocycloaddition or Michael addition. Based on this hypothesis, we used L-cysteine (L-Cys, a mimic for cytochathiazine A synthesis, Fig. 1c) and adenine (a mimic for alachalasin F synthesis, Fig. 1c) as the donors, under acidic conditions (in pH 4 Tris-HCl buffer), taking the Michael addition reaction with **7** as an example. Apart from the product **2**, the corresponding Michael addition products **9** and **10** were successfully detected by LC-MS (Fig. 4c, i, ii, and Fig. 3a), and further confirmed by high-resolution mass spectrometry (HRMS) (Supplementary Figs. 23, 24). These results strongly indicate that the previous reported pcCYTs and meCYTs are possibly not natural products, but instead, they are likely artificially derived products, which mainly depend on the reactive promiscuity of the γ-keto-α,β-unsaturated moiety in the macrocycle framework of aliphatic amino acid-type moCYTs.

**Berberine bridge enzyme (BBE)-like oxidase AspoA alters the native and nonenzymatic pathways.** We next investigated the function of the flavin-dependent oxidase gene *aspoA*. AspoA contains a berberine bridge enzyme/glycolate oxidase (BBE/GlcD) conserved domain (Supplementary Fig. 9a) and belongs to the BBE-like oxidase superfamily[30]. BBE-like oxidases usually catalyse dehydrogenation or dehydrogenation-mediated C-C or C-N bond formation reactions during natural product biosynthesis[31–35]. In many *cyt* BGCs, a gene which is homologous to the flavin-dependent oxidase *aspoA* replaces the presence of a gene encoding a BVMO (Supplementary Fig. 2). In contrast to *AN-aspoEHBCF*, the strain *AN-aspoEHBCFA* produced two new compounds, **11** (~0.5 mg/L, aspochalasin Q) and **12** (~0.7 mg/L, aspochalasin P), with the same molecular weights as compounds **7** and **8** (Fig. 5a, i, ii), which indicated that unlike the classical BBE-like oxidase, AspoA does not catalyse dehydrogenation reactions of **7** or **8**. Large-batch fermentation and isolation of **11** and **12** (Supplementary Tables 10, 11 and Supplementary Figs. 71–83) showed the following: (1) these compounds are the double bond isomerization counterparts of **7** and **8**, respectively (Fig. 3); (2) the γ-keto-α,β-unsaturated moiety in **7** and **8** is converted to a 1,4-diketone in **11** and **12**, which possibly removes the high reactivity. Indeed, upon further incubation of **11** (as the example substrate) with L-cysteine or adenine in pH 4 Tris-HCl buffer, the expected pcCYTs of **11**, as well as their corresponding Michael addition meCYTs, were not detected (Fig. 5b, i–iii), and compound **11** was stable. These results clearly suggest that AspoA acts as a switch to alter the native and nonenzymatic pathways in aspochalasin synthesis. The actual route to synthesize aspochalasin in *A. flavipes* KLA03 is the avoidance of nonenzymatic conversions, such as intramolecular cyclization to form pcCYTs and intermolecular addition to form meCYTs.

BBE-like oxidases usually have two conserved fingerprint motifs, "R/KxxGH" and "CxxV/L/IG"[36]. His in motif 1 and Cys in motif 2 are the key residues responsible for the unusual bicovalent attachments to the 8α and 6 positions of the isoalloxazine ring of the cofactor FAD[37]. Unlike the identified fungal BBE-like enzymes (such as EasE[34], Supplementary Fig. 9b), AspoA has only the conserved $H_{158}$ residue of motif 1, while the $C_{226}$ residue of motif 2 in AspoA is mutated to $Gly_{226}$ ($G_{226}$, Supplementary Fig. 9b). This spontaneous mutation indicates that

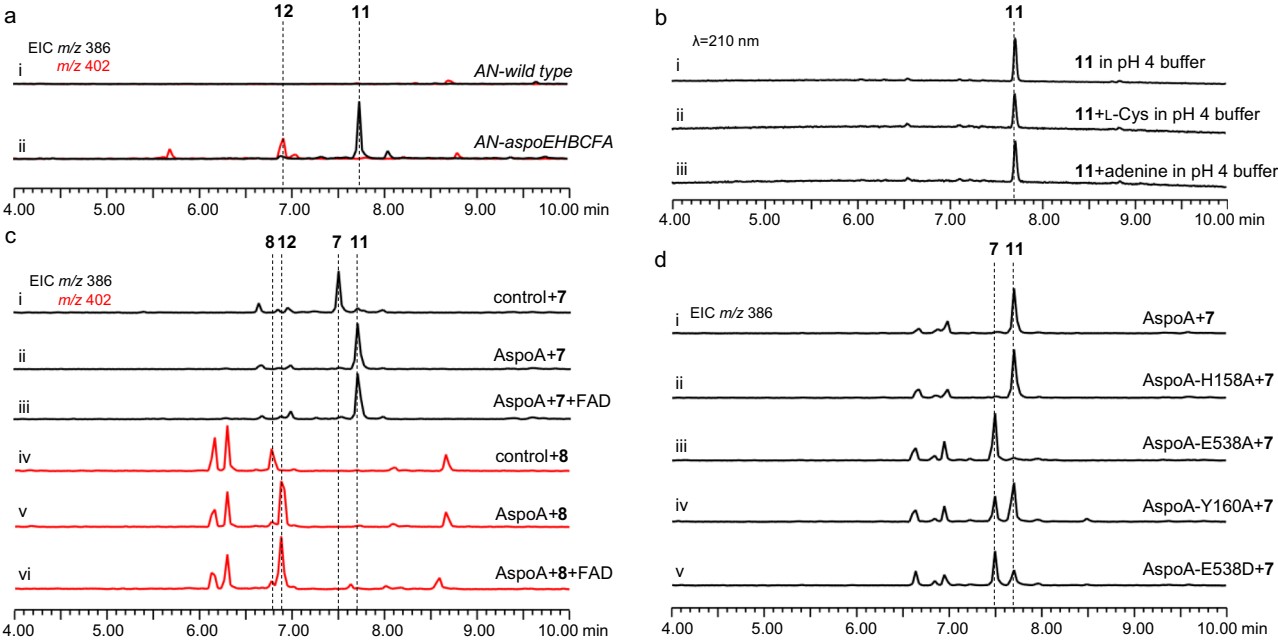

**Fig. 5 Confirmation of the function of gene *aspoA*. a** LC-MS analyses of the culture extracts from the *A. nidulans* transformants. **b** Compound **11** could not undergo nonenzymatic conversions under acidic conditions. **c** In vitro biochemical assays showed that AspoA catalyses the isomerization of **7** or **8** to **11** or **12**, respectively, where the exogenous addition of FAD does not increase the activity of AspoA. **d** Identification of the key amino acid residues in AspoA for double bond isomerization by site-directed mutation. Mutation of the classical endogenous FAD binding residue $His_{158}$ does not decrease the activity of AspoA. Site-direct mutagenesis demonstrated that $Glu_{538}$ is essential for AspoA activity. The EICs were extracted at $m/z$ 386 [M + H]$^+$ for **7** and **11**, $m/z$ 402 [M + H]$^+$ for **8** and **12**.

AspoA has a rare mono-covalent flavin linkage[30]. Phylogenetic analysis and sequence similarity network (SSN) analysis further showed that it is indeed divided into a separate evolutionary clade (Supplementary Fig. 9c, d).

**AspoA uses $Glu_{538}$ as the general acid biocatalyst to catalyse a protonation-driven double bond isomerization reaction.** To confirm the function of AspoA, intron-free *aspoA* was cloned and expressed in *E. coli*; however, soluble expression of AspoA was not successful even when glutathione *S*-transferase (GST)-tagged or maltose binding protein (MBP)-tagged AspoA was constructed (Supplementary Fig. 10a). Alternatively, yeast was used as the heterologous expression host, and the activity of AspoA was then confirmed by cell-free extraction. After incubation of **7** and **8** with AspoA, production of **11** and **12** was detected by LC-MS analysis (Fig. 5c, i, ii, iv, v). Additionally, adding exogenous 100 μM FAD (final concentration) or FMN (Supplementary Fig. 11) did not increase the activity of AspoA (Fig. 5c, iii, vi). Moreover, the $H_{158}A$ mutant (elimination of the endogenous binding ability of AspoA toward FAD or FMN) did not decrease the activity of AspoA (Fig. 5d, i, ii). These two results indicate that the cofactor FAD (FMN), which is essential for the activity of classical BBE-like oxidases, likely does not participate in AspoA-catalysed reaction.

To discover the key amino acid residues and to deduce the mechanism of AspoA, we attempted to use a molecular docking model to investigate the interaction of AspoA with **7** and **8**. A flavoprotein oxidase MtVAO615 (PDB 6F72)[38], with known crystal structure reported, from *Myceliophthora thermophila* C1 was found via homologue modelling of the Swiss Model online analysis[39]. Although the function of MtVAO615 has not been identified, the MtVAO615 shows 35.15% identity with AspoA, which meets the requirement of accuracy of docking model that the identity between the template and the query enzyme should exceed 30%(Supplementary Fig. 12)[40]. As shown in Fig. 6a, b, the

molecular docking models show that there are indeed no alkaline amino acid residues (such as His, Arg or Lys, as observed for classical BBE-like oxidases to catalyse the dehydrogenation reaction) to act as the base to abstract the $C_{18}$ hydrogen, however, this indicates another possible mechanism of AspoA (Fig. 3b). (1) The proposed distance between $Glu_{538}$ and the $C_{21}$ carbonyl group is 3.2 Å (for **7**) and 2.5 Å (for **8**) (Fig. 6c, d), highly indicating that proton transfer to the $C_{21}$ carbonyl group from $Glu_{538}$ is possible;[41] (2) protonation of the $C_{21}$ carbonyl group could promote the $C_{19}$-$C_{20}$ double bond shift and remove the $C_{18}$ hydrogen; (3) during this process, $Tyr_{160}$ possibly stabilizes the $C_{18}$ hydroxyl group via a hydrogen bonding interaction (the proposed distance between $Tyr_{160}$ and the $C_{18}$ hydroxyl group) is 2.8 Å (for **7**) and 2.7 Å (for **8**) (Fig. 6c, d) or as the base pair (in enzymatic environment) to abstract the $C_{18}$ hydrogen[42] (Supplementary Fig. 13); and (4) the final keto-enol tautomerization of both $C_{21}$ and the $C_{18}$ carbonyl group furnishes the conversion of **7** to **11** or **8** to **12**. Notably, the AspoA homologous proteins and the predicted key residues $Glu_{538}$ and $Tyr_{160}$ are highly identical and conserved in many aliphatic amino acid-type *cyt* BGCs (Supplementary Fig. 9b).

Based on this hypothesis, we carried out a series of site-direct mutation experiments, using **7** as the example substrate to compare with the AspoA wild type, and found that (1) the $E_{538}A$ mutant indeed abolished the activity of AspoA (Fig. 5d, iii); (2) the $Y_{160}A$ mutant decreased the activity of AspoA (Fig. 5d, iv); and (3) importantly, the $E_{538}D$ mutant retained the ability to catalyse the conversion of **7** to **11** (Fig. 5d, v), which fully suggests the role of $Glu_{538}$ as the general acid biocatalyst that catalyses the protonation of the $C_{21}$ carbonyl group. The decrease in activity of the $E_{538}D$ mutant may be due to Glu having one additional methylene unit, which may position the acidic side chain close to the substrate carbonyl group (the proposed distance between the $Asp_{538}$ mutant and the $C_{21}$ carbonyl group of **7** and **8** increased to 3.9 Å and 3.3 Å, respectively, Fig. 6e, f).

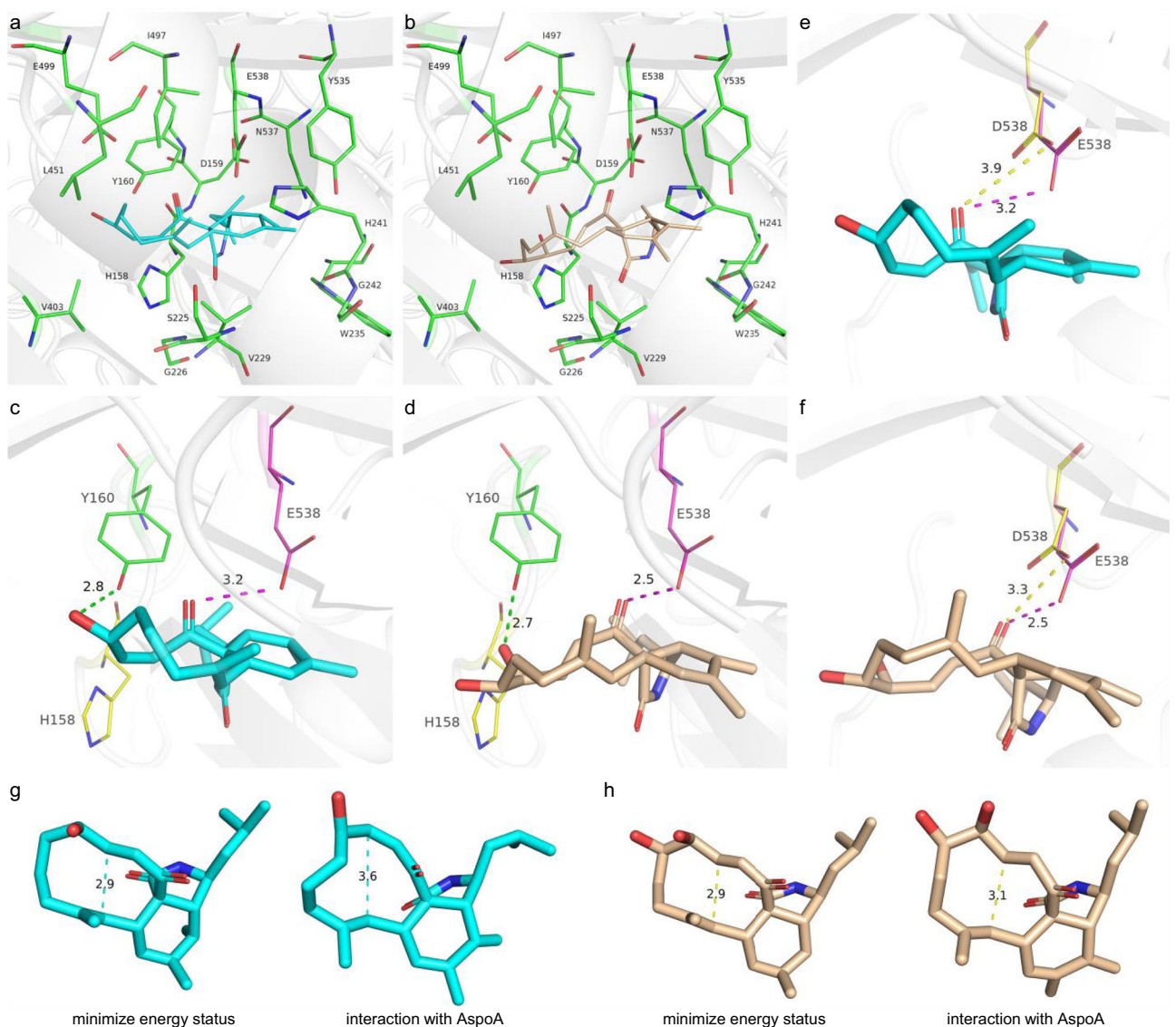

**Fig. 6 Molecular docking model of AspoA with substrates 7 and 8.** The proposed complex structures of AspoA with **7** (**a**) and **8** (**b**). The proposed distance between AspoA-Glu$_{538}$ and the C$_{21}$ carbonyl group of **7** (**c**) and **8** (**d**), or the AspoA-Tyr$_{160}$ and C$_{18}$ hydroxy group of **7** (**c**) and **8** (**d**). The proposed distance between AspoA-Glu$_{538}$ or AspoA mutant Asp$_{538}$ and C$_{21}$ carbonyl group of **7** (**e**) and **8** (**f**). The proposed distance between C$_{13}$ and C$_{19}$ of **7** (**g**) and **8** (**h**) in minimize energy status (left) or in interaction with AspoA status (right). Compounds **7** and **8** are shown in cyan and wheat colours, respectively.

To confirm the mechanism of AspoA as shown in Fig. 3b, we carried out the enzymatic reaction in D$_2$O buffer (SI) and found the following information. (1) When the purified **11** (*m/z* 386 [M + H]$^+$) was incubated in D$_2$O buffer, the molecule weight of **11** does not increase, which confirmed that the hydrogen-deuterium exchange in **11** cannot be occurred (Supplementary Fig. 14a–c). However, (2) when the AspoA-catalyzed isomerization of **7** to form **11** was instead performed in D$_2$O buffer, the molecule weight of the generated **11** increased by 2 amu (*m/z* 388 [M + H]$^+$, Supplementary Fig. 14a–c), highly suggesting the proposed dienol intermediate is indeed exist (Fig. 3b). (3) When the enzyme-prepared $^2$H-**11** (*m/z* 388 [M + H]$^+$) was incubated back to H$_2$O buffer, the molecule weight of the $^2$H-**11** does not decrease (Supplementary Fig. 14a–c), which confirmed that these two deuteriums were incorporated into the nonactivated carbon atoms of **11**, respectively (Supplementary Fig. 14c, e). (4) The $^2$H-**11** was finally prepared from the large-scale enzymatic conversion

assays (SI), and the subsequent $^1$H NMR analysis showed that these two deuteriums were indeed incorporated into C$_{19}$ and C$_{20}$ of **11** (Supplementary Fig. 14d, e), respectively. (5) The spontaneous conversion of **7** to **2** in pH 4 D$_2$O buffer confirmed that only one deuterium was incorporated into C$_{20}$, while the incorporated deuterium was also not further wash-out during incubation of $^2$H-**2** back to H$_2$O buffer (Supplementary Fig. 15a–e). The above both amino acid residues mutation and isotope labelling results confirmed that the AspoA-catalysed double bond isomerization includes protonation of the C$_{21}$ carbonyl group, hydride shift and keto-enol tautomerization (Fig. 3b and Supplementary Fig. 14e).

Although these two conversions use the same precursors (**7** and **8**) and are all achieved via protonation of the C$_{21}$ carbonyl group (Fig. 3b), compared to the nonenzymatic conversion to form **2** and **1**, AspoA strictly catalyses the production of **11** and **12**. These results clearly suggest that the C$_{13}$-C$_{14}$ double bond, as the nucleophile to form the new C$_{13}$-C$_{19}$ bond, should be

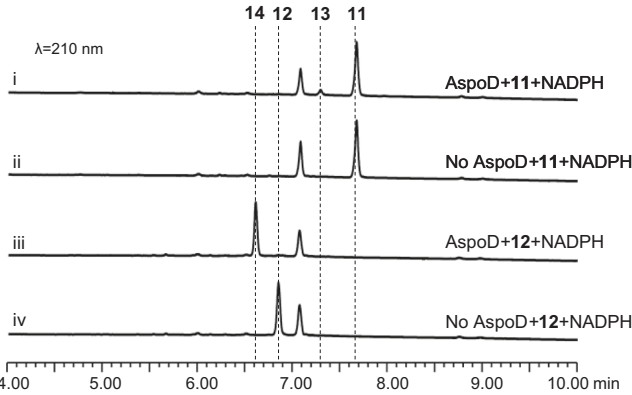

**Fig. 7 Confirmation of the function of AspoD.** In vitro biochemical assays showed that AspoD catalyses the reduction of **11** or **12** to **13** or **14**, respectively.

completely inhibited in the AspoA-catalysed reaction. However, the possible corresponding aromatic amnio acid residues of AspoA that are used for the stabilization of the $C_{13}$-$C_{14}$ double bond via π-π interactions were not found (Fig. 6a, b). Therefore, we reasoned that during the AspoA-catalysed reaction, **7** and **8** may have slightly different conformations. Indeed, as shown in Fig. 6g, h, the distance between $C_{13}$ and $C_{19}$ (minimized energy status *vs.* interaction with AspoA status) increased from 2.9 Å (for **7**) and 2.9 Å (for **8**) to 3.6 Å (for **7**) and 3.1 Å (for **8**), respectively, which may prevent $C_{13}$-$C_{19}$ bond formation.

**AspoD acts as the cooperation partner of AspoA, and it specifically and precisely reduces the $C_{18}$ carbonyl group.** The function of the last gene, *aspoD*, was confirmed by in vitro biochemical assays. N-His6 AspoD was expressed and purified from *E. coli* (Supplementary Fig. 10b). When AspoD was incubated with **11** and **12** in the presence of NADPH, two corresponding reduction products, **13** ($m/z$ 388 $[M + H]^+$) and **14** ($m/z$ 404 $[M + H]^+$, flavichalasine G, Supplementary Table 12 and Supplementary Fig. 84–90), were detected (Fig. 7, i–iv). However, the $k_{cat}/K_m$ calculation showed that the catalytic efficiency of AspoD towards **12** was nearly 15-fold greater than that of **11** (Supplementary Fig. 16), which indicates that **12** is the preferred substrate of AspoD.

These results fully demonstrate that AspoD acts as the cooperation partner of AspoA, it specifically and precisely reduces the $C_{18}$ carbonyl group (formed by AspoA-catalysed double bond isomerization and subsequent keto-enol tautomerization) back to its original hydroxyl group (Fig. 3a). For instance, for the conversions of **7** to **13** or **8** to **14**, via this isomerization, tautomerization and reduction strategy (catalysed by AspoA-AspoD together), the key $C_{19}$-$C_{20}$ double bond in **7** or **8** was skillfully eliminated; however, other functional groups in **7** or **8** were not changed. This directional conversion ensures the metabolic flux in the direction of the native pathway of aspochalasin, while the nonenzymatic conversions are blocked.

In this work, we clarified the gene function of the *aspo* cluster and successfully reconstituted the core backbone as well as the whole pathway of the cytochalasin family compounds. Significantly, the flavin-dependent oxidase AspoA harbours the BBE-like oxidase feature but uses $Glu_{538}$ as the general acid biocatalyst, which catalyses an unusual protonation-driven double bond isomerization reaction and finally alters the native and nonenzymatic pathways in aspochalasin synthesis. Our results greatly promote research progress on the heterologous biosynthesis of cytochalasin family compounds, importantly present an unprecedented function of BBE-like enzymes in natural product

biosynthesis and highly suggest that the isolated pcCYTs and meCYTs are most likely artificially derived products.

## Method

**General methods**. Reagents were purchased from Sigma-Aldrich, Thermo Fisher Scientific, or New England BioLabs. Primer synthesis and DNA sequencing were performed by Sangon Biotech Co., Ltd. (Shanghai, China). The plasmids and primers used in this study are summarized in Supplementary Tables 1–3. All plasmids were extracted by the alkaline lysis method and dissolved in elution buffer. LC-MS analyses were performed on a Waters ACQUITY H-Class UPLC-MS system coupled to a PDA detector and an SQD2 mass spectrometer (MS) detector with an ESI source. Chromatographic separation was performed at 35 °C using a C18 column (ACQUITY UPLC® BEH, 1.7 μm, 2.1 mm × 100 mm, Waters). MPLC was performed on BUCHI Reveleris® X2 Flash Chromatography System, with UV and ELSD detectors using BUCHI Reveleris® C18 column (40 μm, 80 g). Semi-preparative HPLC was performed on Shimadzu Prominence HPLC system using a YMC-Pack ODS-A column (5 μm, 10 × 250 mm). MCI column chromatography (CC) was performed on an MCI gel CHP 20 P/120 (37–75 μm, Mitsubishi Chemical Corporation, Japan). NMR spectra were recorded on a Bruker AVANCE III NMR (400 MHz) with a 5 mm broadband probe and TMS as an internal standard. HRMS data were obtained on Fourier-transform ion cyclotron resonance-mass spectrometry (FT-ICR-MS) (Bruker SolariII, Bremen, Germany) or quadrupole time-of-flight (QTOF) mass spectrometry (Bruker IMPACT II, Bremen, Germany).

**Strains**. *Aspergillus flavipes* KLA03 was cultured on PDB medium (26 g/L Potato Dextrose Water) at 25 °C for 4 days for extraction of genomic DNA (gDNA) and complementary DNA (cDNA). *Aspergillus nidulans* LO8030 was used as the host for heterologous expression of the *aspo* gene cluster. *Saccharomyces cerevisiae* strain BJ5464-NpgA was used as the host for the expression of *aspoA* or for heterologous recombination to construct the *A. nidulans* overexpression plasmids. *Escherichia coli* BL21 was used for protein expression of *aspoA* and *aspoD*. *E. coli* XL-1 was used for cloning.

**Isolation of the gDNA and cDNA synthesis**. *A. flavipes* KLA03 was cultivated in PDB medium at 25 °C for 4 days to extract gDNA according to cetyl-trimethylammonium bromide (CTAB) methods and to extract RNA by TRLZOL® Reagent (Ambion). RNA samples were then treated with DNase, followed by cDNA reverse transcription with the Transcriptor First Strand cDNA Synthesis Kit (Roche).

**The preparation and transformation of *A. nidulans* protoplasts**. *A. nidulans* was cultured in solid CD medium (10 g/L glucose, 50 mL/L 20 × nitrate salts, 1 mL/L trace elements, 20 g/L agar) containing 10 mM uridine, 5 mM uracil, 1 μg/mL pyridoxine HCl and 0.25 μg/mL riboflavin at 37 °C for 5 days, and then spores were collected in 20% glycerol. The spores were inoculated in 40 mL liquid CD medium and cultured at 37 °C and 220 rpm for 9 h. After the germination of spores, culture fluid was centrifuged at 4 °C, 2000 g for 5 min to harvest the mycelia. The precipitation was washed two times with 15 mL Osmotic buffer (1.2 M MgSO₄·7H₂O, 10 mM sodium phosphate, pH 5.8) and centrifuged at 4 °C and 2000 g for 5 min to remove the supernatant. Then, the precipitate was resuspended in 10 mL osmotic buffer containing 30 mg Lysing Enzymes (Sigma) and 20 mg Yatalase (Takara), transferred into a 50 mL Erlenmeyer flask, and cultured at 28 °C and 80 rpm for 14 h. The culture fluid was poured directly into a sterile 50 mL centrifugal tube and overlaid gently with 10 mL of trapping buffer (0.6 M sorbitol, 0.1 M Tris-HCl, pH 7.0), and then centrifuged at 4 °C and 3000 g for 20 min. The protoplasm layer was transferred and fully scattered into 2xSTC buffer (1.2 M sorbitol, 10 mM CaCl₂, 10 mM Tris-HCl, pH 7.5), and centrifuged at 4 °C and 3000 g for 8 min. The supernatant was removed, and STC buffer was added to resuspend the protoplasts for transformation.

**Heterologous expression of the *aspo* cluster in *A. nidulans***. To gain stains of heterologous expression in *A. nidulans*, 2–5 μL plasmids (pIM 8001–8007) were added to 100 μL protoplasts of *A. nidulans* and held on ice for 30 min. Subsequently, 600 μL PEG solution was added into the mixture and the mixture was cultured on the regeneration dropout solid medium (CD medium with 1.2 mM sorbitol and appropriate supplements, CD-SD medium) at 37 °C after being placed at room temperature for 20 min. After 2-3 days, the transformants were moved on solid CD and cultivated at 37 °C for 3–4 days to for sporulation. The spores were inoculated on solid CD-ST medium (20 g/L starch, 10 g/L casein hydrolysate (acid), 50 mL/L nitrate salts, 1 mL/L trace elements, 20 g/L agar) and cultured at 25 °C for 3 days, while the products were analysed using LC-MS.

**Metabolite analysis for *A. nidulans* strains**. The transformant of *A. nidulans* was grown on solid CD-ST for 3 days and extracted with ethyl acetate. The organic phases were evaporated to dryness and dissolved in methanol for LC-MS analyses. LC-MS metabolite profiles were performed on a Waters UPLC-MS system with the following method: chromatographic separation was achieved with a linear gradient

of 5–99% MeCN-H$_2$O (both with 0.02% v/v formic acid) in 10 min followed by 99% MeCN for 3 min and then 5% MeCN-H$_2$O for 3 min, with a flow rate of 0.4 mL/min. The MS data were collected in the $m/z$ range 50–1500 in positive mode simultaneously.

**Feeding assays of [1,2-$^{13}$C]-L-leucine in *A. nidulans*.** 1 mM [1,2-$^{13}$C]-L-leucine (final concentration) was added to 4 ml solid CD-ST medium and the spores of *AN-aspoEH* and *AN-aspoEHB* were inoculated on medium. Then the petri dishes were maintained at 25 °C for 3 days, and products were extracted with a twofold volume of ethyl acetate. The extracted ethyl acetate layer was evaporated to dryness, redissolved in methanol, and then analyzed by LC-MS.

**Feeding assays of 6 and 7 for AspoF in *A. nidulans*.** The recombinant plasmid pIM8006 was transformed into *A. nidulans* to obtain strain *AN-aspoF*. The strain was cultured in 40 ml liquid CD-ST medium at 25 °C, 220 rpm for 2.5 days and then centrifugated to remove all solution. The cells were resuspended in 3 mL liquid CD-ST medium and cultured at 25 °C and 220 rpm for 12 h after 200 μM substrate (compound **6** or **7**) was added. The products were extracted with twofold volume of ethyl acetate. The extracted ethyl acetate layer was evaporated to dryness, redissolved in methanol, and then analyzed by LC-MS.

**The protein expression and purification of AspoD in *E. coli*.** To confirm the function of the *aspoD* gene, AspoD protein was expressed and purified from *E. coli*. The recombinant plasmid pIM 8011 was transformed into the *E. coli* BL21 strain by heat shock transformation. The mono colony was cultivated in 3 ml liquid LB medium (25 g/L LB broth) with 100 μg/mL ampicillin at 37 °C overnight. The bacterial solution was then transferred to 300 mL LB medium containing 100 μg/mL ampicillin and cultured at 37 °C and 220 rpm to an OD$_{600}$ of 0.4–0.6. Then, the cells were maintained at 16 °C for 30 min and cultured at 16 °C for 20 h after 0.2 mM isopropylthio-β-D-galactoside (IPTG) was added. After that, the cells were collected by centrifugation at 4 °C and 3000 g for 5 min and resuspended in 15 mL buffer A (50 mM Tris-HCl, 500 mM NaCl, 10% glycerol, pH 7.5). Subsequently, the cells were lysed through sonication on ice and centrifuged at 4 °C and 23,000 g for 40 min to gain the soluble fraction. The protein was purified by Ni-NTA agarose resin and the protein of interest was eluted by buffer A containing 350 mM imidazole. The purified protein was passed through a PD-10 desalting column (GE Healthcare) and eluted with buffer C (50 mM Tris-HCl, 50 mM NaCl, 5% glycerol, pH 7.5). The protein was concentrated using a 30-kDa ultrafiltration centrifugal tube (Millipore Amicon ® Ultra-15 mL) at 4 °C and 2000 g. The concentrated protein solutions were aliquoted into 1.5 ml EP tubes, flash frozen with liquid nitrogen, and then stored at −80 °C. The purified enzyme was analysed by SDS-PAGE, and the concentration was measured with a BCA protein quantification kit (Beijing Dingguo Changsheng Biotechnology Co., Ltd).

**In vitro characterization of AspoD.** An in vitro assay for AspoD was performed in 50 μL buffer C (pH 7.5), containing 5 μM AspoD, 400 μM NADPH and 200 μM substrate (compound **11** or **12**). The reaction was quenched with an equal volume of MeOH after 2 h of incubation at 25 °C, and centrifuged at 23,000 g for 5 min before LC-MS analysis.

**In vitro characterization of AspoA and its mutants.** Plasmids pIM8012-8016 were transformed into the heterologous expression host *S. cerevisiae* through the Frozen-EZ Yeast Transformation II Kit (Zymo Research) and the transformant yeast strains were selected on solid selective uracil dropout medium at 28 °C for 2–3 days and confirmed by colony PCR. The right single colony was inoculated into 3 mL liquid uracil dropout medium and cultured at 28 °C and 220 rpm for 15–24 h. The inoculums were then inoculated on YPD medium (20 g/L glucose, 20 g/L tryptone, 10 g/L yeast extract) and cultured at 28 °C and 250 rpm for 48 h. The culture broth was centrifuged to remove solution and collect the cells. The cells were lysed by grinding, and cellular debris was resuspended in buffer C. The supernatant was harvested by centrifugation at 4 °C and 23,000 g for 30 min. Substrates were added in supernatant to accomplish the reaction at 25 °C for 10 h. The final concentration of compounds **7** or **8** and the cofactors FAD or FMN were 100 μM. The reaction mixture was extracted with twice the volume of ethyl acetate. The extracted ethyl acetate layer was evaporated to dryness and redissolved in methanol. The products were analysed by LC-MS.

**The nonenzymatic reactions in Tris-HCl buffer.** All spontaneous reactions were performed in 200 μL pH 4 Tris-HCl buffer, 100 μM compounds **7** and **8** were added and the reactions were incubated at 25 °C for 10 h to obtain **2** and **1**. For mimic synthesis of meCYT products (**9** and **10**), 100 μM compound **7** and 200 μM L-cystine or adenine were added. After 6 h at 25 °C, the reactions were quenched and extracted with 200 μL of ethyl acetate. The resultant organic extracts were evaporated to dryness, redissolved in methanol, and then analysed by LC-MS.

**Molecular docking.** The amino acid sequence of AspoA was submitted to SWISS-MODEL (https://swissmodel.expasy.org/). Based on sequence similarity, the crystal structure of MtVAO615 from *Myceliophthora thermophila* (PDB: 6F72,

https://www.rcsb.org/structure/6F72) was selected as the template to model the tertiary structure of AspoA. Molecular docking between the AspoA model and substrates **7** or **8** was carried out using the software Discovery studio client (Discovery Studio V2021, National Demonstration Center for Experimental Pharmacy Education, Southwest University) by means of the CDOCKER method. The interrelation between the AspoA model and substrates was analysed by Discovery studio client.

## Data availability

The sequence data of *aspo* gene cluster from *A. flavipes* KLA03 is listed in Supplementary information. All other data generated and analyzed in this study are available within the article and the Supplementary information. Source data are provided with this paper.

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

## Acknowledgements
We thank Prof. Mancheng Tang from Shanghai Jiaotong University and Prof. Xiao Zhang from Southwest University, for their helpful discussions. This work is supported by the National Key R&D Program of China (2020YFA0907700 to Y.Z.) and Chongqing Science Funds for Distinguished Young Scientists (cstc2020jcyj-jqX0005 to Y.Z.).

## Author contributions
J-M.Z. and X.L. performed all in vivo and in vitro experiments, as well as compounds isolation. Q.W. performed compounds characterization. C.T.M and J-M.Z. performed molecular docking. J-M.Z., X.L., Q.W., C.T.M., D.H.L. and Y.Z. analysed and discussed the results. Y.Z. conducted the research and wrote the manuscript.

## Competing interests
The authors declare no competing interests.
