## [Peer Review File · Nature Communications]

REVIEWER COMMENTS

Reviewer #1 (Remarks to the Author):

In this manuscript, Zou and coworkers characterized the gene function of the aspo gene cluster for a cytochalasin compound. Authors successfully reconstituted the cytochalasin carbon backbone, as well as the whole pathway in a heterologous host *Aspergillus nidulans*. During this reconstitution, authors clarified the processing steps and identified key enzymes in the biosynthesis of Aspo including Diels-Alderase, BBE-like oxidase, etc. One of the most fascinating part of this work is a BBE-like enzyme, which catalyses an unusual protonation-driven double bond isomerization reaction.

-The structure of 3 is quite complex. Yet only one sentence of the manuscript "Isolation of 3 from the large-batch fermentation cultures of AN-aspoEH was carried out... Supplementary Fig. 38-43). I would have hoped to see a more thorough discussion of the elucidation process to obtain confidence in the proposed structures.

-Authors stated hydrolase (AspoC) would influence the efficiency of Diels-Alderase (AspoB), but why? Did these enzymes have protein-protein interaction. What is the exact function of hydrolase. Authors should give more discussions.

-Authors claimed the FAD/FMN dose not participate in the reaction of BBE-like enzyme on the basis that adding 100 uM FAD did not increase the activity of AspoA. However, in many FAD-dependent enzymes, the cofactor FAD is tightly binding to enzyme, with or without adding additional FAD did not affect the activity.

-Authors used a molecular docking model to investigate the molecular basis of AspoA catalyzed reaction. The accuracy of docking model is highly dependent on the template enzyme. Authors should give more details of the template enzyme.

-Author propose the mechanism of double bond shift and ether bond formation, the reviewer will encourage authors to carry out the enzymatic reaction in deuterium buffer to support their hypothesis.

-Authors claimed the catalytic efficiency of AspoD toward 12 was nearly 20-fold greater than that of 11. It would be better if author calculate the k_{cat}/K_m of these enzymes.

Based on the above-described points, I believe that there are major issues to be addressed before this manuscript could be considered acceptable for publication.

Reviewer #2 (Remarks to the Author):

Key results:

This paper significantly expands our understanding of cytochalasin biosynthesis through heterologous expression experiments, in vitro studies, and cell free reactions. The authors have identified areas where knowledge of specific steps in cytochalasin biosynthesis is lacking, and have attempted to address these areas with studies into the biosynthesis of the aspochalasins. They also present the first heterologous production of the whole pathway to a cytochalasin in a heterologous host. Finally they present studies on the berberine bridge enzyme (BBE)-like oxidase AspoA which they propose catalyses an unusual protonation-driven double bond isomerisation reaction.

Validity:

I believe this paper presents a robust set of experiments regarding the biosynthesis of cytochalasins. I do not think there are any significant flaws that would prohibit publication.

Significance:

This is a highly significant paper, that presents the first successful heterologous productions of the whole pathway to a cytochalasin in a heterologous host. Indeed, despite several attempts by other groups, this study represents the first reconstruction of a core backbone to a cytochalasin in a host organism. They also determine that the existence of the more complex mero-cytochalasins and polycyclic-cytochalasins are likely the product of non-enzymatic reactions. This is shown to be the

case for the investigated pathway to the aspochalasans. Here they also demonstrate that the unusual BBE-like oxidase AspoA prevents conversion of the mono-cytochalasans 7 and 8 to mero-cytochalasans and polycyclic-cytochalasans through an unusual isomerisation reaction.

Data and methodology:

The approaches used appear to be valid. Reconstitution of the aspochalasans pathway in a heterologous host is demonstrated through expression of sequential sets of genes involved in the pathway, in experiments which are well controlled. In vitro experiments complement this data, along with mutations to specific residues to allow mechanistic proposals. These experiments are also well controlled. I do not have expertise in the areas of protein modelling or NMR, so I cannot comment on the validity of the data for these sections.

Clarity and context:

The paper is clear in its objectives, and the outcomes of the various experiments. I believe it has been presented with sufficient context and consideration of previous work.

Suggested improvements:

The paper is mostly very well written, and the experiments are comprehensive. I have several suggestions for improvements:

1. I'm not sure exactly what the authors mean by 'gradually' in the following sentence:
The structural complexity of CYTs is mainly attributed to four gradually variable bioconversion processes
2. Should the word 'oxidative' be replaced with 'oxidised' in the following sentence?
tailoring steps that are catalyzed by numerous distinctive oxidases to form highly oxidative functional groups
3. In Scheme 1, an indication of what the colour scheme represents would be useful. Also the red colour seems to represent both the isoindole core and other chemical moieties in the formation of mero-cytochalasans. It would be clearer if these were represented by two different colours.
4. The following sentence seems to suggest that the four-gene conserved cassette of cyt BGCs always synthesises the same initial core backbone:
Identification of the initial core backbone synthesized by the four-gene conserved cassette (consisting of PKS-NRPS, trans-ER, hydrolase and the Diels-Alderase, Supplementary Fig. 2) of all the cyt BGCs.
I think the authors are trying to say that the conserved cassette is present in all cyt BGCs and they want to identify a specific core backbone, so perhaps it would be clearer to say:
Identification of an initial core backbone synthesized by the four-gene conserved cassette (consisting of PKS-NRPS, trans-ER, hydrolase and the Diels-Alderase, Supplementary Fig. 2) which is common to all cyt BGCs.
5. 'Amino' is spelt incorrectly in the following sentence:
Reconstitution of aromatic ammino acid-type cyt BGCs...
6. The following sentence might read better replacing 'Except for' with 'Apart from':
Except for the four-gene conserved cassette (aspoEHBC) and regulation gene (aspoG), cluster 2 (aspo cluster) has three tailoring genes...
7. Regarding the addition of the hydrolase gene aspoC (to produce AN-aspoEHBC) which increased the yield of 6 in comparison to strain AN-aspoEHB, how was this increase quantified? Were the fermentations and extractions directly comparable e.g. inoculated with the same number of spores etc? And was the AN-aspoEHBC strain directly generated from the AN-aspoEHB strain?
8. I'm not clear on the meaning of the following sentence, could this be re-worded / expanded:
Starting the Diels-Alder reaction mainly depends on the formation of a hydrolase-bound intermediate to retain the correct tautomer form of the substrate.
9. The following sentence should replace 'previously' with 'previous':
This discovery is the opposite of a previously biosynthetic hypothesis, that formation of polycyclic...
10. I'm not clear in the following sentence if this hypothesis is being proposed by the authors, or has previously been proposed. If it has been previously proposed, this should be referenced:

This nonenzymatic polycyclic transformation has been proposed to be related to the highly reactive features of the γ -keto- α,β -unsaturated moiety in 7 and 8...

11. The following sentence would be clearer if 'Except for' is replaced with 'Apart from'
Except for the product 2, the corresponding Michael addition products 9 and 10...

12. This sentence is a bit unclear:

The homologous enzymes of AspA are conserved and take the place of BVMOs in many cyt clusters

If I'm correct in interpreting what the authors are trying to say, it might be better re-worded as the following:

In many cyt BGCs a gene which is homologous to the flavin-dependent oxidase *aspA* replaces the presence of a gene encoding a BVMO.

13. This sentence is missing a full stop:

The actual route to synthesize aspochalasin in *A. flavipes* KLA03 is the avoidance of nonenzymatic conversions, such as intramolecular cyclization to form pcCYTs and intermolecular addition to form meCYTs

14. This sentence could be reworded:

Phylogenetic tree analysis further showed that it is indeed divided into a separated evolution clade. It might sound better:

Phylogenetic analysis further showed that it is indeed divided into a separate evolutionary clade.

15. Regarding the phylogenetic analysis, it would be useful to have the accession numbers for each of the BBE-like oxidases used in the phylogenetic tree, either in the figure, or as a separate table.

16. The authors mention that the lack of conservation of the cysteine residue in motif 2, places the cyt BBE-like oxidases in the atypical type II BBE-like oxidase family. However, since the production of the H158A mutant (motif 1) has no impact on activity, this suggests that AspA is not in the rare mono-covalent flavin linkage type II BBE-like oxidase family at all, since AspA does not appear to require flavin to function, and a lack of conservation of either motif 1 or 2 does not impact activity. Are any of the type II BBE-like oxidases included in the phylogenetic analysis? It would be interesting to see if they clade with the cyt BBE-like oxidases which would more definitively place them in this family. I suggest that the authors either do a more comprehensive phylogenetic tree, to see which family the cyt BBE-like oxidases fall into, or if they are a separate family altogether, or be more speculative when talking about which family these are in.

17. The molecular docking studies are not my area of expertise, but I wondered if more information could be provided on how accurate the model is likely to be? E.g. how similar is the template to AspA? How closely does the model match to the tertiary structure of the template?

18. The methods section requires some editing to improve the English.

Reviewer #3 (Remarks to the Author):

Cytochalasins are fungal polyketide-nonribosomal peptide hybrid. Its backbone biosynthesis has been well established in recent years. The manuscript entitled "Berberine bridge enzyme-like oxidase-catalyzed double bond isomerization acts as the pathway switch in cytochalasin synthesis" by Zhang et al. initially focuses on the biosynthesis of polycyclic and polymerized cytochalasin. During the characterization of polycyclic CYT, a BBE-like enzymes catalyzed double bond isomerization in the cluster was elucidated. The principal findings of this manuscript: 1) The reconstitution of the cytochalasin core backbone as well as the whole pathway in a heterologous host *Aspergillus nidulans*. 2) The polycyclic cytochalasins 1 and 2 are not the enzyme-catalyzed products from the biosynthetic process of the aspochalasin. They are the products acid-mediated nonenzymatic cyclization of compounds 7 and 8, respectively. 3) Identification and in vitro testing of a berberine bridge enzyme-like oxidase AspA catalyzed the double bond isomerization. 4) Molecular docking study suggested that AspA used Glu 538 as the general acid to push a

protonation-driven double bond isomerization reaction. Based on these results, the authors propose a reasonable biosynthetic pathway for compounds 1, 2, 13 and 14.

Characterization of a new family enzyme-catalyzed new chemical reaction is generally important for scientific communities. However, their finding of the function of AspA in this manuscript is not so significant, because the enzymatic reaction mechanism contains just common keto-enol isomerization and hydride shift and thus easily presumed. If authors can provide an enzyme crystal structural analysis with the bonded substrate to elucidate the detailed mechanism instead of the simulated structure, the manuscript would be improved. Following are some questions and major concerns need to be addressed.

1. The conclusion about the pcCYTs and meCYTs are not the enzyme-catalyzed products from the biosynthetic process of the aspochalasin family of compounds is not accurate. The author only demonstrates two polycyclic CYT 1 and 2 are yielded by spontaneous reaction. This can not be expanded to other pcCYTs, even to polymerized CYT.

2. The authors suggest that Tyr160 acts as an acid to interact with the C18 hydroxy group. The mechanism is chemically incorrect. How about acts as a base pair?

3. The authors need draw a detailed mechanism for the hydride shift during the double bond isomerization and provide experimental evidences. The isotope labeling study (labeled water) is suggested.

4. I'm curious about the data present in the main manuscript. It seems that the amount of compound 8 in figure 3C-iv is much lower than those of in figures 3C-v-vi? The peak of 8 in figure 3C-iv should theoretically be higher or stronger.

Overall, I don't recommend the publication of current manuscript in Nat Commun.

Review 1 Comments	Reply/Changes
I believe that there are major issues to be addressed before this manuscript could be considered acceptable for publication.	Thank you. We have addressed each comment and a point-by-point reply is listed below.
The reviewer hopes to see a more thorough discussion of the elucidation process for compound 3 to obtain confidence in the proposed structures.	Thank you. The elucidation process for compound 3 is listed below. Compound 3 was isolated as a yellow powder. HRMS gave an exact mass of 368.2585 for [M - H]⁻ (calcd. 368.2595 for C₂₄H₃₄NO₂), suggesting a molecular formula of C₂₄H₃₅NO₂ and indicated eight degrees of unsaturation. The ¹H NMR spectrum showed the presence of an amide NH (δ_H 7.49, br s), seven olefinic protons [δ_H 5.53 (1H, o), 5.56 (1H, s), 5.88 (1H, d), 6.17 (2H, d), 6.32 (1H, dd), 6.79 (1H, d)], three methyl doublets (δ_H 0.94×2, and 1.74), two methyl singlets (δ_H 1.78×2), and 11 methine and/or methylene protons. Detailed analyses of ¹³C NMR, DEPT and HSQC spectra revealed that it has five methyl, five methylene, eight methine, and five quaternary carbons in addition to an amide carbonyl (δ_C 172.6). Four of the quaternary carbons were olefinic (δ_C 107.0, 134.0, 135.7, 137.7), and one was oxygenated (δ_C 162.4). Besides seven degrees of unsaturation occupied by carbonyls and double bonds (δ_C 97.8, 121.7, 122.5, 125.8, 126.4, 135.2, 143.1, 172.6), the remaining suggested compound 3 possessing one ring system. Analysis of the ¹H-¹H COSY NMR data of compound 3 identified the C-3'/C-4'/C-5'/C-6', C-4/C-5/C-6/C-7/C-8/C-9, C-11/C-12/C-13 and C-15/C-16 four moieties. HMBC correlations from Me-17 to C-9/C-10, Me-18 to C-13/C-14, from olefinic proton H-11 and H-15 to olefinic quaternary carbon C-10 and C-14, and H-4 to oxygenated quaternary carbon C-3 established the long chain substructure. A Five membered lactam ring was confirmed by HMBC correlation from olefinic proton H-2' to olefinic quaternary carbon C-1'/C-2 and to amide carbonyl C-1. Isobutyl group was adjacent to C-1' supported by HMBC correlations among H-3' between C-1'. The double bonds configuration of compound 3 was determined based on ¹H-¹H coupling constants and NOESY experiments. The coupling constants of J_{4,5} = 15.4 Hz and J_{12,13} = 15.3 Hz and the NOESY correlations of H-12 with Me-17 and Me-18, H-11 with H-13 indicated the presence of four E-configured double bonds. Therefore, the structure of compound 3 was determined. The elucidation process for compound 3 is described in SI.
The reviewer recommends the authors to give more	Thank you for your helpful suggestion. Determination

discussions about the exact function of hydrolase.	of the exact function of hydrolase in R domain reductive release-type PKS-NRPS cluster (including aspo cluster in this study) is a very difficult work because the possible native intermediate of hydrolase has never been identified or captured. Recently, using synthesized mimic substrates, Prof. Cox's group proposed that the hydrolase needs first covalent binding the intermediate to retain the correct tautomer form of the substrate, which is essential for the followed Diels-Alder reaction during pyrichalasin H biosynthesis (Chemistry, 2021, 27, 3106-3113). In our work, (1) without hydrolase AspoC, the Diels-Alderase AspoB could catalyse the cyclization of the proposed native intermediate 4 to form 6; however, (2) addition of AspoC significantly increased the yield of 6 (almost 60%, Fig. 1b, vi). These results suggested that hydrolase AspoC only influences but does not determine the catalytic ability of AspoB during aspochalasin biosynthesis. Based on these results, we proposed that the hydrolase AspoC possibly provides a structural cavity (not via covalent binding) to retain 4 in the correct tautomer form to react with Diels-Alderase AspoB (via inhibit the spontaneous conversion of 4 to 3 as far as possible). We added this discussion in main text, however, the exact function of hydrolase AspoC needs further deep investigation.
Authors claimed the FAD/FMN dose not participate in the reaction of BBE-like enzyme on the basis that adding 100 uM FAD did not increase the activity of AspoA. However, in many FAD-dependent enzymes, the cofactor FAD is tightly binding to enzyme, with or without adding additional FAD did not affect the activity.	Thank you. Indeed, the endogenous binding FAD/FMN is important for the classical flavin oxidases. BBE-like oxidases usually have two conserved fingerprint motifs, "R/KxxGH" and "CxxV/L/IG". His in motif 1 and Cys in motif 2 are the key residues responsible for the unusual bi-covalent attachments to the 8α and 6 positions of the isoalloxazine ring of the cofactor FAD/FMN (J. Biol. Chem., 2016, 281, 21276-21285). Unlike the identified fungal BBE-like enzymes (such as EasE, J. Am. Chem. Soc., 2019, 141, 17517-17521, Supplementary Fig. 9b), AspoA has only the conserved H₁₅₈ residue of motif 1, while the C₂₂₆ residue of motif 2 in AspoA is mutated to Gly₂₂₆ (G₂₂₆, Supplementary Fig. 9b). This spontaneous mutation indicates that AspoA is a FAD/FMN mono-covalent binding flavin oxidase. As shown in Fig. 3d, when the H₁₅₈ of AspoA was mutated (elimination of the endogenous binding ability of AspoA toward FAD/FMN), the His₁₅₈A mutant retained the same isomerase activity. Moreover, addition of 100 uM FAD/FMN did not increase the activity of AspoA. Therefore, the cofactor FAD/FMN does not participate in AspoA-catalyzed isomerization reaction was confirmed by evidence both pro and con.
The reviewer suggests the authors to give more details of the template enzyme for the molecular docking model.	Thank you for your helpful suggestion. We first searched the homologue modelling of AspoA by the Swiss Model (https://swissmodel.expasy.org). A flavoprotein MtVAO615 (PDB 6f73.1, Molecules 2018, 23, 111) with unknown function from Myceliophthora thermophila C1 was found. MtVAO615 has 35.15% identity toward AspoA, which meets the requirement of accuracy of docking model that the identity between the template and the query enzyme should exceed 30% (Protein Sci. 2006, 15, 808-824). The molecular docking of AspoA with substrates 7 and 8 were carried out using the software Discovery studio client (2021, National Demonstration Center for Experimental

	Pharmacy Education, Southwest University) by means of CDOCKER method. The interrelation between AspA model and substrates were analyzed by Discovery studio client as well. We added these details in main text.
The reviewer encourages the authors to carry out the enzymatic reaction in deuterium buffer to support the mechanism of double bond shift and ether bond formation.	Thank you for your helpful suggestion. Using compound 7 as the example substrate, we carried out the enzymatic reaction of AspA as well as the spontaneous conversions in D₂O buffer to further confirm our hypothesis on AspA-catalysed unusual protonation-driven double bond isomerization. The following results are listed below: (1) When the purified 11 (m/z 386 [M+H]⁺) was incubated in D₂O buffer, the molecule weight of 11 does not increase, which confirmed that the hydrogen-deuterium exchange in 11 cannot be occurred (Supplementary Fig. 13a-c). However, (2) when the AspA-catalyzed isomerization of 7 to form 11 was instead performed in D₂O buffer, the molecule weight of the generated 11 increased by 2 amu (m/z 388 [M+H]⁺, Supplementary Fig. 13a-c), highly suggesting the proposed dienol intermediate is indeed exist (Scheme 2b). (3) When the enzyme-prepared ²H-11 (m/z 388 [M+H]⁺) was incubated back to H₂O buffer, the molecule weight of the ²H-11 does not decrease (Supplementary Fig. 13a-c), which confirmed that these two deuteriums were incorporated into the nonactivated carbon atoms of 11, respectively (Supplementary Fig. 13e). (4) The ²H-11 was finally prepared from the large-scale enzymatic conversion assays (SI), and the subsequent ¹H NMR analysis showed that two deuteriums were indeed incorporated into C₁₉ and C₂₀ of 11 (Supplementary Fig. 13d-e), respectively. (5) The spontaneous conversion of 7 to 2 in pH 4 D₂O buffer confirmed that only one deuterium was incorporated into C₂₀, while the introduced deuterium was also not further wash-out during incubation of ²H-2 back to H₂O buffer (Supplementary Fig. 14a-e). Therefore, combination of the AspA mutation with the deuterium-labeled results, the mechanism of the AspA-catalysed unusual protonation-driven double bond isomerization was demonstrated. We added the brief descriptions of deuterium-labeled results in main text and added two new Supplementary Figs. 13 and 14 in SI.
The reviewer recommends the authors calculate the kcat/km of AspD toward 12 and 11 to better characterize the catalytic efficiency.	Thank you for your helpful suggestion. As shown in new Supplementary Fig. 15, the kinetic parameters of AspD toward 12 and 11 were measured, respectively. Based on these results, the catalytic efficiency of AspD toward 12 was 15-fold greater than that of 11 .
Review 2 Comments	Reply/Changes
I believe this paper presents a robust set of experiments regarding the biosynthesis of cytochalasans. I do not think there are any significant flaws that would prohibit publication.	Thank you.
The reviewer is not sure exactly what the authors mean by 'gradually' in the following sentence: "the structural complexity of CYTs is mainly attributed to four gradually variable bioconversion processes".	Thank you. We deleted the word "gradually", which makes the sentence to understand clearly.
The reviewer suggests that the word 'oxidative' should	Thank you for your helpful suggestion. Changed as

be replaced with 'oxidised' in the sentence "tailoring steps that are catalyzed by numerous distinctive oxidases to form highly oxidative functional groups".	suggested.
In Scheme 1, an indication of what the colour scheme represents would be useful. Also the red colour seems to represent both the isoindole core and other chemical moieties in the formation of mero-cytochalasans. It would be clearer if these were represented by two different colours.	Thank you for your helpful suggestion. As shown in new Scheme 1, we used red colour to represent the isoindole core and purple colour to represent other chemical moieties in the formation of mero-cytochalasans.
The sentences "Identification of the initial core backbone synthesized by the four-gene conserved cassette (consisting of PKS-NRPS, trans-ER, hydrolase and the Diels-Alderase, Supplementary Fig. 2) of all the cyt BGCs." is replaced with "Identification of an initial core backbone synthesized by the four-gene conserved cassette (consisting of PKS-NRPS, trans-ER, hydrolase and the Diels-Alderase, Supplementary Fig. 2) which is common to all cyt BGCs." That would be clearer.	Thank you for your helpful suggestion. Changed as suggested.
'Amino' is spelt incorrectly in the following sentence: "Reconstitution of aromatic ammino acid-type cyt BGCs..."	Thank you. Changed as suggested.
The following sentence might read better replacing 'Except for' with 'Apart from': "Except for the four-gene conserved cassette (aspoEHBC) and regulation gene (aspoG), cluster 2 (aspo cluster) has three tailoring genes..."	Thank you for your helpful suggestion. Changed as suggested.
Regarding the addition of the hydrolase gene aspoC (to produce AN- aspoEHBC) which increased the yield of 6 in comparison to strain AN- aspoEHB , how was this increase quantified? Were the fermentations and extractions directly comparable e.g. inoculated with the same number of spores etc? And was the AN- aspoEHBC strain directly generated from the AN- aspoEHB strain?	Thank you. (1) All the fermentation and extraction experiments of AN- aspoEHBC and AN- aspoEHB were undergone at the same conditions including the same amounts of spores, the same fermentation hours, the same fermentation centigrade, and the same way of extraction by organic solvents. The production of 6 was quantified via the peak area integral by LC-MS analysis. (2) To introduce aspoC but to avoid introducing one more plasmid, we added the gene aspoC into the aspoHB plasmid (pIM 8003) to create the new expression aspoHBC plasmid (pIM 8004). The combination of aspoE+aspoHB or aspoE+aspoHBC were transferred into A. nidulans to generate the AN- aspoEHB strain and the AN- aspoEHBC strain, respectively.
The reviewer is not clear on the meaning of the following sentence, could this be re-worded / expanded: "Starting the Diels-Alder reaction mainly depends on the formation of a hydrolase-bound intermediate to retain the correct tautomer form of the substrate."	Thank you for your helpful suggestion. Changed as suggested.
The following sentence should replace 'previously' with 'previous': "This discovery is the opposite of a previously biosynthetic hypothesis, that formation of polycyclic..."	Thank you for your helpful suggestion. Changed as suggested.
The reviewer is not clear in the following sentence if this hypothesis is being proposed by the authors, or has previously been proposed. If it has been previously proposed, this should be referenced: "This nonenzymatic polycyclic transformation has been proposed to be related to the highly reactive features of the γ -keto- α,β -unsaturated moiety in 7 and 8 ..."	Thank you. This hypothesis is proposed by the authors. Re-worded as the following: "This nonenzymatic polycyclic transformation might be related to the highly reactive features of the keto- α,β -unsaturated moiety in 7 and 8 ..."
The following sentence would be clearer if 'Except for' is replaced with 'Apart from': "Except for the product 2 , the corresponding Michael addition products 9 and	Thank you for your helpful suggestion. Changed as suggested.

10..."	
This sentence is a bit unclear: "The homologous enzymes of AspA are conserved and take the place of BVMOs in many cyt clusters". It might be better re-worded as the following: "In many cyt BGCs a gene which is homologous to the flavin-dependent oxidase aspoA replaces the presence of a gene encoding a BVMO."	Thank you for your helpful suggestion. Changed as suggested.
This sentence is missing a full stop: "The actual route to synthesize aspoA in A. flavipes KLA03 is the avoidance of nonenzymatic conversions, such as intramolecular cyclization to form pcCYTs and intermolecular addition to form meCYTs"	Thank you. We added a full stop.
The sentence "Phylogenetic tree analysis further showed that it is indeed divided into a separated evolution clade." could be reworded: "Phylogenetic analysis further showed that it is indeed divided into a separate evolutionary clade."	Thank you for your helpful suggestion. Changed as suggested.
Regarding the phylogenetic analysis, it would be useful to have the accession numbers for each of the BBE-like oxidases used in the phylogenetic tree, either in the figure, or as a separate table.	Thank you for your helpful suggestion. We added NCBI accession number for each BBE-like oxidase in phylogenetic tree.
Are any of the type II BBE-like oxidases included in the phylogenetic analysis? The reviewer suggests that the authors either do a more comprehensive phylogenetic tree, to see which family the cyt BBE-like oxidases fall into, or if they are a separate family altogether, or be more speculative when talking about which family these are in.	Thank you for your helpful suggestion. To find more cyt BBE-like oxidases from the fungal genome database, we used the sequence similarity network (SSN) method to construct a more comprehensive phylogenetic tree. As shown in Supplementary Fig. 9d, AspA and its homologue proteins are indeed in a separate family clade, which not only indicates the unique function of them, but also shows the wide distribution in fungi.
The reviewer recommends the authors provide more information on how accurate the model is likely to be? E.g. how similar is the template to AspA? How closely does the model match to the tertiary structure of the template?	Thank you for your helpful suggestion. We first searched the homologue modelling of AspA by the Swiss Model (https://swissmodel.expasy.org). A flavoprotein MtVAO615 (PDB 6f73.1, Molecules 2018, 23, 111) with unknown function from Myceliophthora thermophila C1 was found. MtVAO615 has 35.15% identity toward AspA, which meets the requirement of accuracy of docking model that the identity between the template and the query enzyme should exceed 30% (Protein Sci. 2006, 15, 808-824). The molecular docking of AspA with substrates 7 and 8 were carried out using the software Discovery studio client (2021, National Demonstration Center for Experimental Pharmacy Education, Southwest University) by means of CDocker method. The interrelation between AspA model and substrates were analyzed by Discovery studio client as well. We added these details in main text.
The methods section requires some editing to improve the English.	Thank you for your suggestion. We polished the language in methods section by professional services.
Review 3 Comments	Reply/Changes
However, their finding of the function of AspA in this manuscript is not so significant, because the enzymatic reaction mechanism contains just common keto-enol isomerization and hydride shift and thus easily presumed.	BBE-like oxidases are wide distribution in bacterial and fungal genome. This family enzymes usually catalyse dehydrogenation or dehydrogenation-mediated C-C or C-N bond formation reactions during natural product biosynthesis. However, in this work, the BBE-like oxidase AspA uses Glu ₅₃₈ as the general acid biocatalyst to catalyze an unusual protonation-driven double bond isomerization reaction. This new function of AspA has not been reported in BBE-like oxidases. Therefore, our finding is important, it presents an unprecedented function of BBE-like enzymes in natural

	product biosynthesis and expands the potential application of it in chemoenzymatic synthesis.
The reviewer recommends the authors provide an enzyme crystal structural analysis with the bonded substrate to elucidate the detailed mechanism instead of the simulated structure.	Thank you. Crystal structure of AspoA is hampered, because the soluble expression of AspoA (or with codon optimization) in E.coli is not successful even when Glutathione S-transferase (GST)-tagged or Maltose binding protein (MBP)-tagged AspoA was constructed. Alternatively, yeast was used as the heterologous expression host, and the activity of AspoA was successfully confirmed by cell-free extraction. Combination of the molecular docking, key amino acid residues mutation as well as the deuterium-labeled experiments, the catalytic mechanism of AspoA was demonstrated.
The conclusion about the pcCYTs and meCYTs are not the enzyme-catalyzed products from the biosynthetic process of the aspochalasin family of compounds is not accurate. The author only demonstrates two polycyclic CYT 1 and 2 are yielded by spontaneous reaction. This can not be expanded to other pcCYTs, even to polymerized CYT.	Thank you. Based on our both in vivo and in vitro results, the pcCYTs and meCYTs of aspochalasin family compounds and their structural analogues indeed originate from nonenzymatic pathway. Moreover, other nonenzyme-synthesized pcCYTs or meCYTs has been observed recently via chemical synthesis (J. Org. Chem. 2017, 82, 9704-9709; Angew. Chem. Int. Ed. 2018, 57, 14221-14224; Angew. Chem. Int. Ed. 2021, 60, 15963-15971). The main highlight of this study is to identify the new function of BBE-like oxidase AspoA, and to demonstrate that AspoA catalyses unusual double bond isomerization reaction and acts as the important pathway switch to alter the native pathway and nonenzymatic pathway in synthesis of cytochalasin compounds. Therefore, AspoA and its homologue proteins in various cyt cluster are the key point in structural diversity of cytochalasins.
The authors suggest that Tyr160 acts as an acid to interact with the C ₁₈ hydroxy group. The mechanism is chemically incorrect. How about acts as a base pair?	Thank you. As shown in Fig. 3d, iv, the Y160A mutant decreased but not eliminated the activity of AspoA. Therefore, based on this result, we proposed that the Tyr160 possibly stabilizes the C ₁₈ hydroxyl group via a hydrogen bonding interaction, which might promote the C ₁₉ -C ₂₀ double bond shift and remove the C ₁₈ hydrogen. Of course, in enzymatic environment, the possibility of Y160 as the base pair to abstract the C ₁₈ hydrogen is not excluded, we added this mechanism in Supplementary Fig. 12.
The authors need draw a detailed mechanism for the hydride shift during the double bond isomerization and provide experimental evidences. The isotope labeling study (labeled water) is suggested.	Thank you for your helpful suggestion. Using compound 7 as the example substrate, we carried out the enzymatic reaction of AspoA as well as the spontaneous conversions in D ₂ O buffer to further confirm our hypothesis on AspoA-catalysed unusual protonation-driven double bond isomerization. The following results are listed below: (1) When the purified 11 (m/z 386 [M+H] ⁺) was incubated in D ₂ O buffer, the molecule weight of 11 does not increase, which confirmed that the hydrogen-deuterium exchange in 11 cannot be occurred (Supplementary Fig. 13a-c). However, (2) when the AspoA-catalyzed isomerization of 7 to form 11 was instead performed in D ₂ O buffer, the molecule weight of the generated 11 increased by 2 amu (m/z 388 [M+H] ⁺ , Supplementary Fig. 13a-c), highly suggesting the proposed dienol intermediate is indeed exist (Scheme 2b). (3) When the enzyme-prepared ² H- 11 (m/z 388 [M+H] ⁺) was incubated back to H ₂ O buffer, the

	molecule weight of the ^2H-11 does not decrease (Supplementary Fig. 13a-c), which confirmed that these two deuteriums were incorporated into the nonactivated carbon atoms of 11, respectively (Supplementary Fig. 13e). (4) The ^2H-11 was finally prepared from the large-scale enzymatic conversion assays (SI), and the subsequent ^1H NMR analysis showed that two deuteriums were indeed incorporated into C₁₉ and C₂₀ of 11 (Supplementary Fig. 13d-e), respectively. (5) The spontaneous conversion of 7 to 2 in pH 4 D₂O buffer confirmed that only one deuterium was incorporated into C₂₀, while the introduced deuterium was also not further wash-out during incubation of ^2H-2 back to H₂O buffer (Supplementary Fig. 14a-e). Therefore, combination of the AspoA mutation with the deuterium-labeled results, the mechanism of the AspoA-catalysed unusual protonation-driven double bond isomerization was demonstrated. We added the brief descriptions of deuterium-labeled results in main text and added two new Supplementary Figs. 13 and 14 in SI.
It seems that the amount of compound 8 in figure 3C-iv is much lower than those of in figures 3C-v-vi? The peak of 8 in figure 3C-iv should theoretically be higher or stronger.	Thank you. We used the same concentration of compound 8 to perform AspoA cell-free extracts assays and control assays. In three independent experiments (Figure R1), as shown below, the final extraction peak of substrate 8 (in AspoA-free control) indeed lower than the generated product 12 (in AspoA reaction assay) by LC-MS analysis. Therefore, the difference in ionization level of compound 8 and 12 under our analysis condition is the possible reason.

Figure R1. Three independent experiments that AspoA-catalysed isomerization of **8** to **12**.

REVIEWERS' COMMENTS

Reviewer #1 (Remarks to the Author):

The reviewer is satisfied with the changes. This manuscript is now acceptable.

Reviewer #2 (Remarks to the Author):

I would like to thank the authors for their responses to my comments. In the main, I feel that they have addressed most of my concerns.

Two points remain at least partially unanswered – firstly, I'm still not convinced that 'AspoA belongs to the atypical type II BBE-like oxidase subfamily'. The referenced paper for this sentence states the following 'Interestingly, the occurrence of the active site type II and III appears to be restricted to Brassicaceae.'⁽¹⁾ Therefore it seems quite a jump to state that AspoA belongs to this very specific plant family based only on the mono-covalent flavin linkage. Perhaps the authors could remove the reference to the type II BBE-like subfamily, unless they can provide further evidence. It would also be useful to have a more detailed description in the ESI of exactly how the SSN was made and what it is showing.

Secondly, perhaps the ESI could show an overlay of the template MtVAO615, with the generated model of AspoA, to provide evidence of its reliability as a model.

(1) Daniel, B.; Konrad, B.; Toplak, M.; Lahham, M.; Messenlehner, J.; Winkler, A.; Macheroux, P. The family of berberine bridge enzyme-like enzymes: A treasure-trove of oxidative reactions. Archives of Biochemistry and Biophysics 2017, 632, 88-103. DOI: 10.1016/j.abb.2017.06.023.

Reviewer #3 (Remarks to the Author):

The overall revision improved the manuscript.

I still have one suggestion for this manuscript.

The HRMS for deuterium labeled compounds should be included in the main text and SI. This is quite important for structure identification for compounds without NMR data.

Review 1 Comments	Reply/Changes
The reviewer is satisfied with the changes. This manuscript is now acceptable.	Thank you.
Review 2 Comments	Reply/Changes
Perhaps the authors could remove the reference to the type II BBE-like subfamily, unless they can provide further evidence. It would also be useful to have a more detailed description in the ESI of exactly how the SSN was made and what it is showing.	Thank you for your helpful suggestion. (1) We deleted the sentence “in contrast to the classical BBE-like oxidase, and AspoA belongs to the atypical type II BBE-like oxidase subfamily” and the Cited ref here. (2) We added more detailed description about SSN in Supplementary Figure 9 legend.
The reviewer suggests that the ESI could show an overlay of the template MtVAO615, with the generated model of AspoA, to provide evidence of its reliability as a model.	Thank you for helpful suggestion. We added the overlay of the template MtVAO615 with the generated model of AspoA as the new Supplementary Figure 12 to provide evidence of its reliability as a model.
Review 3 Comments	Reply/Changes
The overall revision improved the manuscript.	Thank you.
The reviewer suggests that the HRMS for deuterium labeled compounds should be included in the main text and SI.	Thank you for helpful suggestion. We supplement HRMS for deuterium labeled compounds as new Supplementary Figures 28-29.